# 5-HT_1A_ Receptor Agonist Treatment Partially Ameliorates Rett Syndrome Phenotypes in *mecp2*-Null Mice by Rescuing Impairment of Neuron Transmission and the CREB/BDNF Signaling Pathway

**DOI:** 10.3390/ijms232214025

**Published:** 2022-11-14

**Authors:** Hongmei Dai, Yoshikazu Kitami, Yu-ichi Goto, Masayuki Itoh

**Affiliations:** Department of Mental Retardation and Birth Defect Research, National Institute of Neuroscience, National Center of Neurology and Psychiatry, Kodaira, Tokyo 187-8551, Japan

**Keywords:** Rett syndrome, tandospirone, 5-HT_1A_ receptor, neurotransmitter

## Abstract

Rett syndrome (RTT) is an X-linked neurodevelopmental disorder caused by mutations in the gene that encodes methyl CpG-binding protein 2 (MECP2) and is characterized by the loss of acquired motor and language skills, stereotypic movements, respiratory abnormalities and autistic features. There has been no effective treatment for this disorder until now. In this study, we used a *Mecp2*-null (KO) mouse model of RTT to investigate whether repeated intraperitoneal treatment with the 5-HT_1A_ receptor agonist tandospirone could improve the RTT phenotype. The results showed that administration of tandospirone significantly extended the lifespan of *Mecp2*-KO mice and obviously ameliorated RTT phenotypes, including general condition, hindlimb clasping, gait, tremor and breathing in *Mecp2*-KO mice. Tandospirone treatment significantly improved the impairment in GABAergic, glutaminergic, dopaminergic and serotoninergic neurotransmission in the brainstem of *Mecp2*-KO mice. Decreased dopaminergic neurotransmission in the cerebellum of *Mecp2*-KO mice was also significantly increased by tandospirone treatment. Moreover, RNA-sequencing analysis found that tandospirone modulates the RTT phenotype, partially through the CREB1/BDNF signaling pathway in *Mecp2*-KO mice. These findings provide a new option for clinical treatment.

## 1. Introduction

Rett syndrome (RTT) is an X-linked neurodevelopmental disorder that affects approximately 1 in 10,000 girls. RTT patients are characterized by seemingly normal development up to 6–18 months of age and then display a sudden deceleration in growth and development, which are associated with the progressive loss of acquired motor and language skills, stereotypic hand movements, muscle hypotonia, autonomic dysfunctions and severe cognitive impairment [1]. Most RTT cases are caused by mutations in the gene coding for methyl CpG-binding protein 2 (MECP2) [2]. MeCP2 is most abundant in the central nervous system (CNS), associated with neuronal maturation and synaptogenesis [3,4,5]. MeCP2 deficiency in mice results in RTT-like symptoms consisting of hind limb clasping, tremors, breathing irregularities, loss of muscle tone and hypoactivity [6]. Defects in various neurotransmitter systems have been observed in patients and animal models of RTT syndrome [7,8]. Currently, there is no curative therapy for RTT, despite many clinical trials and basic research studies investigating RTT.

Serotonin (5-HT) is a modulatory neurotransmitter involved in a variety of physiological and behavioral functions, including anxiety, aggressive behavior, stress, blood pressure regulation, heart rate and the coagulation system. The cellular actions of serotonin are mediated by multiple receptors, which have been classified as 5-HT 1A-F, 5-HT 2A-C, 5-HT 3, 5-HT 4, 5-HT 5, 5-HT 6 and 5-HT 7, according to their structural and functional characteristics [9]. Among all of the 5-HT receptors, the 5-HT 1A receptor (5-HT_1A_) is the most widespread, existing both pre- and postsynaptically [10]. In general, activation of these presynaptic 5-HT_1A_ autoreceptors hyperpolarizes the cell membrane and results in a reduction in the firing rate of serotonergic neurons in the raphe area, leading to suppressed serotonin synthesis, turnover and release. The activation of 5-HT_1A_ receptors on postsynaptic cells decreases the firing rate of postsynaptic cells [10,11]. In mammals, dysfunction of the serotonergic system has been linked to various neurological diseases, such as depression, schizophrenia, Alzheimer’s disease, Parkinson’s disease and autism spectrum disorder [12,13,14,15,16]. Additionally, disturbances in serotoninergic neurotransmission were also found in RTT patients and mouse models [7,17].

Tandospirone is a potent and selective 5-HT_1A_ partial agonist that also presents a high selectivity of two to three orders of magnitude over dopamine, adrenergic and other 5-HT receptors [18]. Tandospirone is an effective anxiolytic drug that is well tolerated and presents limited adverse effects and low abuse liability [19]. Improvement in motor dysfunction in Parkinson’s disease and cognitive deficits in schizophrenia have been revealed in recent clinical and preclinical studies [20,21]. For an RTT patient, tandspirone administration was described to clinically recover respiratory condition and behavior [22]. However, the effect of tandospirone was only demonstrated in one care report. In this study, we verified whether repeated intraperitoneal treatment with tandospirone could improve the RTT phenotype in a mouse model. Here, we revealed that administration of tandospirone was able to improve RTT phenotypes, including general condition, hindlimb clasping, gait, tremor and breathing, which are related with the improvement in GABAergic, glutaminergic, dopaminergic and serotoninergic neurotransmission. Moreover, RNA-seq analysis found that tandospirone modulates the RTT phenotype partially through the CREB1/BDNF signaling pathway in Mecp2-KO mice. 

## 2. Results

### 2.1. Effect of Tandospirone Treatment on Lifespan, Body Weight and Pathological Phenotypes of Mecp2-KO Mice

*Mecp2*-KO mice manifest a progressive postnatal neurological phenotype. *Mecp2*-KO mice are apparently normal until 4 weeks of age and then develop RTT-like symptoms, including inertia, irregular breathing, abnormal gait and hindlimb clasping. The progression of symptoms in *Mecp2*-KO male mice leads to rapid weight loss and a shortened lifespan (death at ~10 weeks of age) compared with WT mice [6,23]. Therefore, we started tandospirone treatment when the animals were 4 weeks of age. After 4 weeks of treatment, at 8 weeks of age, both lifespan and body weight were observed. The survival curve for the treatment of *Mecp2*-KO mice is shown in Appendix A. Statistical analysis indicated that tandospirone treatment significantly extended the lifespan of *Mecp2*-KO mice (50% longer, *p* < 0.001, Appendix A). We measured body weight at 8 weeks of age. The body weight of *Mecp2*-KO mice was significantly lower than that of WT littermates (Appendix A, *p* < 0.001), but there was no significant difference between vehicle and tandospirone treatment. These findings suggest that tandospirone treatment potentially extends the lifespan of *Mecp2*-KO mice but has no effect on the body weight.

Next, we analyzed the effect of tandospirone treatment in the *Mecp2*-KO mice on the pathological phenotype of the Rett syndrome animal model. The score test values were measured at 8 weeks of age according to previous reports [6,24], including general condition, hindlimb clasping, tremor, breathing, mobility and gait (Figure 1A–F, respectively). The scores of *Mecp2*-KO mice (vehicle group) were significantly increased compared to WT littermates, but significantly decreased by tandospirone treatment in general condition (Figure 1A), hindlimb clasping (Figure 1B), gait (Figure 1D), tremor (Figure 1E) and breathing (Figure 1F) were significantly decreased by tandspirone treatment, with no effect on mobility (Figure 1C). These results indicate that tandospirone treatment could partially improve Rett syndrome phenotypes, including retraction of the legs, tremor, irregular breathing and difficulties with walking.

### 2.2. Effect of Tandospirone Treatment on Glutamate, GABA, Monoamines and Metabolites in the Brains of Mecp2-KO Mice

It is known that Mecp2 deficit impairs neurotransmitter systems that correspond with neurological phenotypes observed in both Rett patients and *Mecp2*-KO mice [7,25]. Therefore, in this study, we used an HPLC system to investigate the effect of tandospirone treatment on glutamate, GABA, monoamines and their metabolites in the brains of *Mecp2*-KO mice by HPLC system. The data showed that both the GABA concentration (Figure 2) and glutamate concentration (Figure 3) were significantly decreased in all brain regions of *Mecp2*-KO mice compared with WT littermates. Tandospirone treatment significantly improved the GABA concentration in the brainstem of *Mecp2*-KO mice compared to vehicle (Figure 2E, *p* < 0.01), but the differences were not significant in the other brain regions. The glutamate concentration was also significantly increased in the cerebellum and brainstem of *Mecp2*-KO mice when compared with vehicle (Figure 3D,E, respectively; *p* < 0.05). These results suggest that tandospirone treatment may partially improve the deficits of glutaminergic and GABAergic neurotransmission observed in *Mecp2*-KO mice. In addition, we also confirmed the effect of tandospirone treatment on monoaminergic neurotransmission in the brains of *Mecp2*-KO mice, focusing on the dopaminergic and serotoninergic systems. The results indicated that *Mecp2*-KO mice showed a significantly lower concentration of dopamine and its metabolites in the midbrain (Appendix A, *p* < 0.05), cerebellum (Figure 4A,B, *p* < 0.05) and brainstem (Figure 5A,B, *p* < 0.001) than WT littermates, but the differences were not significant in the cortex (Appendix A, *p* > 0.05) or hippocampus (Appendix A, *p* > 0.05). Tandospirone treatment significantly enhanced the dopamine concentrations of *Mecp2*-KO mice in the cerebellum and brainstem compared to vehicle (Figure 4A and Figure 5A, respectively; *p* < 0.05). The DOPAC concentration in the brainstem was also significantly increased by tandospirone treatment (Figure 5B; *p* < 0.05). These results suggest that tandospirone treatment significantly improves the dopaminergic neurotransmission deficits in the cerebellum and brainstem seen in *Mecp2*-KO mice. The serotonin concentration of *Mecp2*-KO mice was significantly decreased in all brain regions when compared with WT (Appendix A, Figure 4C and Figure 5C, *p* < 0.05). The metabolites HVA and 5-HIAA were significantly reduced in the cortex, midbrain and brainstem (Appendix A, Figure 5D,E, respectively; *p* < 0.05). Tandospirone treatment also had a tendency to increase the low contents of serotonin and its metabolites observed in *Mecp2*-KO mice in different brain regions, compared to vehicle but the increase was only significant in the brainstem (Figure 5C,E, respectively; *p* < 0.05), suggesting that tandospirone treatment might ameliorate the impairment of serotoninergic neurotransmission in the brainstem observed in *Mecp2*-KO mice.

Interestingly, tandospirone also recovered the concentrations of noradrenalin and its metabolite (MHPG; 3-methoxy-4-hydroxy-phenylglycol) in the brainstem of *Mecp2*-KO mice (Appendix A). 

### 2.3. Effect of Tandospirone on Cellular Mechanisms in Mecp2-KO Mice 

To study the therapeutic effect and molecular mechanisms of tandospirone in *Mecp2*-KO mice, the cortex, midbrain, cerebellum and brainstem were collected for RNA-seq analysis after 4 weeks of tandospirone treatment. Bioinformatic and statistical approaches were used to identify the differentially expressed genes (DEGs) from the WT, vehicle and tandospirone-treated *Mecp2*-KO groups. In detail, the dysregulated genes in *Mecp2*-KO mice were first identified by comparing WT and vehicle groups. We defined gene upregulation as vehicle/WT ≥ 2 and downregulation as vehicle/WT ≤ 0.5. Next, the expression of these dysregulated genes observed in brains of *Mecp2*-KO mice was further compared with that in tandospirone-treated *Mecp2*-KO mice. The results showed a total of 129 genes with the potential for recovery of dysregulation by tandospirone treatment in four brain regions, of which 111 were upregulated and 18 were downregulated. Finally, these 129 genes were used for functional pathway analysis using the KEGG pathway database. The results indicated that several candidate genes in the midbrain were associated with the adenylyl cyclase-cyclic adenosine monophosphate (cAMP)-cAMP response element-binding (CREB) signaling pathway, AMP-activated protein kinase (AMPK)-CREB signaling pathway or mitogen-activated protein kinase (MAPK) signaling pathway, including *CAMK2D* (calcium/calmodulin-dependent protein kinase type II subunit delta), *PRKAG2* (protein kinase AMP-activated noncatalytic subunit gamma), *CREB1* (cyclic AMP-responsive element-binding protein 1), *BDNF* (brain-derived neurotrophic factor) and *MAPKAPK2* (MAP kinase-activated protein kinase 2). The *GABRA4* (gamma-aminobutyric acid receptor subunit alpha-4) gene is associated with GABAergic synapse transmission. No pathway was found in the cortex, cerebellum or brainstem. Quantitative PCR (qPCR) indicated that the expression of *CAMK2D*, *PRKAG2*, *CREB1*, *BDNF* and *GABRA4* was significantly decreased in *Mecp2*-KO mice in comparison with WT mice but significantly increased in *Mecp2*-KO tandospirone-treated mice when compared with vehicle *Mecp2*-KO mice (Figure 6A–D,F, respectively; *p* < 0.05). There was no significant change in the expression of *MAPKAPK2* (Figure 6E). These results suggested that tandospirone treatment might activate CREB1 by the cAMP and AMPK signaling pathways in the midbrains of *Mecp2*-KO mice, resulting in increased BDNF expression.

## 3. Discussion

In our present study, we investigated the effect of chronic tandospirone treatment on RTT using a mouse model. By comparing the phenotypes of *Mecp2*-KO tandospirone-treated mice with vehicle, we found that tandospirone treatment significantly extended the lifespan of *Mecp2*-KO mice and obviously ameliorated RTT phenotypes, including general condition, hindlimb clasping, gait, tremor and breathing, in *Mecp2*-KO mice; however, there was no effect of tandospirone treatment on body weight or mobility in these mice. Tandospirone treatment also significantly improved the impairment in GABAergic, glutaminergic, dopaminergic and serotoninergic neurotransmission in the brainstem of *Mecp2*-KO mice. Decreased dopaminergic neurotransmission in the cerebellum of *Mecp2*-KO mice was also significantly increased by tandospirone treatment. Moreover, RNA-seq analysis found that tandospirone modulated the RTT phenotype partially through the CREB1/BDNF signaling pathway in our mouse model. 

Respiratory disorders are prominent and one of the most disturbing features of RTT. This phenotype was faithfully mimicked in our mouse model of RTT, suggesting that breathing dysregulation in RTT results from progressive neurochemical dysfunction in the respiratory network, including GABAergic, glutamatergic and monoaminergic neurotransmission [26]. Recently, several studies have reported that administration of a 5-HT_1A_ receptor agonist reduced apneas, improved breathing irregularity and extended survival in a mouse model of RTT [27,28,29]. Our present results were consistent with these previous studies, proving that treatment with a 5-HT_1A_ receptor agonist could significantly improve RTT brainstem dysfunctions, particularly in the GABAergic, glutamatergic and serotoninergic neurotransmissions, responsible for breathing and lifespan. Moreover, tandospirone is not only a selective 5-HT_1A_ receptor agonist but also binds to the dopamine D2 receptor and affects the dopaminergic pathway [30,31]. It has been reported that dopaminergic stimulation by treatment with levodopa and a dopa-decarboxylase inhibitor improved motor activities, such as tremor, hindlimb clasping, gait, mobility and breathing, in a *Mecp2* mouse model [24]. Thus, we consider improvement in the RTT phenotype upon treatment with tandospirone to be partially modulated by the dopaminergic pathway in the cerebellum and brainstem. In addition, tandospirone treatment may recover noradrenergic neuronal function. 

In addition, disruption in the expression of the brain-derived neurotrophic factor (BDNF) gene has been proposed to contribute to the molecular pathogenesis of RTT [32]. Enhancement of brain BDNF levels improved symptoms of a mouse model of RTT, including locomotor activity, motor coordination and lifespan [33]. BNDF expression is regulated by the CREB family, a nuclear transcription factor expressed in all cells in the brain [34]. In stress-maladaptive mice, chronic administration of 5-HT_1A_ receptor agonists may enhance CREB expression, which, in turn, facilitates transcription factor binding to the BDNF promoter domain to stimulate BDNF synthesis [35]. Therefore, in our present study, we consider that tandospirone treatment increased BDNF expression in *Mecp2*-KO mice due to the activation of CREB1 via CAMK2D and PRKAG2 signaling. Increased BDNF expression partially ameliorates the phenotypes of *Mecp2*-KO mice. However, there are some limitations to completely understanding the molecular mechanism from the results. 

In conclusion, our data indicate that chronic administration of tandospirone may partially ameliorate Rett syndrome phenotypes in *Mecp2*-KO mice by rescuing the impairment of neurotransmission and the CREB/BDNF signaling pathway. These findings provide a new option for clinical treatment.

## 4. Materials and Methods

### 4.1. Mecp2 Knockout (KO) Mice and Preparation

*Mecp2* knockout (KO) mice have been reported previously and genotyping was performed as previously described [36]. Littermate male mice were used in the study and divided into three groups: *Mecp2* wild-type saline-treated (WT), *Mecp2*-KO saline-treated (Vehicle) and *Mecp2*-KO tandospirone-treated (Tandospirone). Mice were intraperitoneally (i.p.) injected with tandospirone (T6704; Merck & Co., Kenilworth, NJ, USA) at a dose of 15 mg/kg/day every day from 4 to 8 weeks of age, according to a previous report [37]. Both the WT and vehicle groups were injected with saline as a control. All experiments were performed in accordance with the National Institutes of Health Guide for the Care and Use of Laboratory Animals. Experimental protocols were approved by the Ethical Committee for Animal Experiments at our institute (Approved protocol number; 2020010).

### 4.2. Score Test of Clinical Symptoms

To analyze the effect of tandospirone treatment on the pathological phenotype of Rett syndrome, the “Score test” was evaluated in *Mecp2*-KO mice according to a previous report adopted in the study of the *Mecp2* null mouse model [24,38]. In detail, the pharmacological treatment started at 4 weeks of age. After 4 weeks of treatment, neurological defects in *Mecp2*-KO mice were scored (blind to the experimenter and treatment), focusing on general condition, hindlimb clasping, tremor, breathing, mobility and gait. Each symptom was scored from 0 to 2; 0 corresponds to the symptom being absent or the same as in the WT mouse, 1 corresponds to the symptom being present and 2 corresponds to the symptom being severe. 

(A) General condition: mouse observed for indicators of general well-being such as coat condition, eyes and body stance. 0 = Clean shiny coat, clear eyes and normal stance. 1 = Eyes dull, coat dull/ungroomed and somewhat hunched stance. 2 = Eyes crusted or narrowed, piloerection and hunched posture. 

(B) Hindlimb clasping: mouse observed when suspended by holding base of the tail. 0 = Legs splayed outward. 1 = Hindlimbs are drawn toward each other (without touching) or one leg is drawn into the body. 2 = Both legs are pulled in tightly, either touching each other or touching the body. 

(C) Tremor: mouse observed while standing on the flat palm of the hand. 0 = no tremor. 1 = Intermittent mild tremor. 2 = Continuous tremor or intermittent violent tremor. 

(D) Breathing: movement of the flanks observed while the animal is standing still. 0 = Normal breathing. 1 = Periods of regular breathing interspersed with short periods of more rapid breathing or with pauses in breathing. 2 = Very irregular breathing—gasping or panting. 

(E) Mobility: the mouse is observed when placed on a bench and then handled gently. 0 = WT. 1 = Reduced movement when compared with WT: extended freezing period when first placed on bench and longer periods spent immobile. 2 = No spontaneous movement when placed on the bench; mouse can move in response to a gentle prod or a food pellet placed nearby. 

(F) Gait: 0 = WT. 1 = Hind legs are spread wider than WT when walking or running with reduced pelvic elevation, resulting in a ‘waddling’ gait. 2 = More severe abnormalities: tremor when feet are lifted, walks backward or ‘bunny hops’ by lifting both rear feet at once.

### 4.3. Measurements of Glutamate, GABA, Monoamines and Their Metabolites in the Brain

Mice were anesthetized and the brain regions being studied, the cortex, hippocampus, midbrain, cerebellum and brainstem, were dissected, immediately frozen in liquid nitrogen and stored at −80 °C until assay. A high-performance liquid chromatography (HPLC) system was used to measure the contents of glutamate, GABA, monoamine and their metabolites in each brain region. To measure the contents of glutamate and GABA, the brain samples were weighed and homogenized in 50 vols 0.3 N perchloric acid containing 1 μM 3,4-dihydroxybenzylamine hydrobromide (DHBA) as an internal standard. Then, the homogenate was centrifuged at 12,000 rpm for 10 min at 4 °C. After filtration (0.45 μm), a 1/10 dilution (20 μL) was injected into the HPLC system. Both glutamate and GABA were separated using an HPLC system at 50 °C on a reverse-phase analytical column (Acclaim 120 C18, 3 mm × 150 mm, 3 μm particle diameter, Thermo Fisher Scientific, Waltham, MA, USA) and detected by an electrochemical detector (CoulArray 8-electrode detector, ESA, MA, USA). The column was eluted with 0.1 M phosphate buffer (pH 6.75) containing 20% methanol. All separations were performed at a flow rate of 1.0 mL/min. The measurement of monoamines and metabolites was also carried out as described in our previous study [39]. The following monoamines and their metabolites were measured: dopamine (DA), 3,4-dihydroxyphenylacetic acid (DOPAC), homovanillic acid (HVA), 5-hydroxytryptamine (serotonin, 5-HT) and 5-hydroxyindoleacetic acid (5-HIAA).

### 4.4. Identification of the Signaling Pathway Regulated by Tandospirone Treatment in Mecp2-KO Mice Using RNA Sequencing (RNA-seq)

Mouse brains regions, including the cortex, midbrain, cerebellum and brainstem, in the WT, vehicle and tandspirone groups (*n* = 4 for each group) were dissected. Total RNA for each brain region was extracted using an RNeasy Plus Mini Kit (Qiagen, Valencia, CA, USA) including gDNA eliminator treatment according to the manufacturer’s protocol. RNA-seq was performed as follows. In brief, the library was prepared using a SMART-Seq v4 Ultra Low Input RNA Kit for Sequencing (Clontech laboratories Inc., Mountain View, CA, USA), Nextera XT DNA Library Prep Kit and Nextera XT Index Kit v2 SetA/B/C/D (Illumina Inc., San Diego, CA, USA) and then sequenced on a NovaSeq 6000 system (Illumina). Data analysis was processed by the following software: Genedata Profiler Genome v13.0.11 (Gendata KK, Tokyo, Japan), STAR v2.6.0c [40] and CLC Genomics Workbench v12 (Qiagen). An absolute value of log2 (fold change) ≥ 1 and adjusted *p* < 0.05 were taken as the screening criteria. After filtering, functional pathways were obtained by using the Kyoto Encyclopedia of Genes and Genomes (KEGG; https://www.genome.jp/kegg/kegg_ja.html) pathway database, accessed on 13 June 2021.

### 4.5. Quantitative PCR (qPCR)

To validate the genes that were differentially expressed in RNA-Seq, we performed real-time qPCR. Total RNA from the cortex, midbrain, cerebellum and brainstem (*n* = 4) was extracted using an RNeasy Mini Kit (Qiagen). The cDNA was synthesized using a High-Capacity cDNA Reverse Transcription Kit (Applied Biosystems, Waltham, MA, USA) according to the manufacturer’s instructions. qPCR was measured by a LightCycler 480 Instrument (Roche Applied Science; Upper Bavaria, Germany). The primers used for the amplification of candidate genes are presented in Appendix A. Amplifications were performed in a 384-well optical plate and the thermocycling conditions were 30 s at 96 °C, 30 s at 60 °C and 30 s at 72 °C for 45 cycles. A quantitative analysis was performed using the ΔΔCT method with normalization to GAPDH expression. Data were obtained from three independent qPCR experiments. 

### 4.6. Statistical Analysis

Data are expressed as the mean ± standard error of the mean (SEM). Statistical analysis was performed using one-way analysis of variance (ANOVA) with Tukey’s multiple comparison post hoc test for intergroup comparisons. All results were analyzed using GraphPad 9.0 Prism software. Lifespan data were plotted as Kaplan–Meier survival curves. The results were considered significant for values of * *p* < 0.05, ** *p* < 0.01, *** *p* < 0.001, **** *p* < 0.0001. 

## Figures and Tables

**Figure 1 ijms-23-14025-f001:**
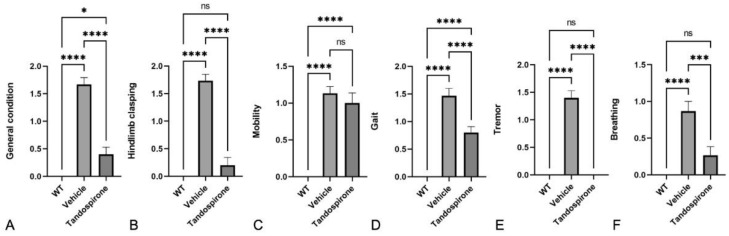
Effect of tandospirone treatment on the *Mecp2*-KO phenotype. Both *Mecp2* WT and *Mecp2*-KO (Vehicle) mice were treated with saline. Score test (*n* = 12–15) related to general condition (**A**), hindlimb clasping (**B**), mobility (**C**), gait (**D**), tremor (**E**) and breathing (**F**). Tandospirone treatment significantly improved the *Mecp2*-KO phenotypes of general condition, hindlimb clasping, gait, tremor and breathing but not in mobility. All values are expressed as the means ± SEMs. Statistical analysis was performed using one-way analysis of variance (ANOVA) with Tukey’s multiple comparison post hoc test for intergroup comparisons. *n* = 12~15 in each test, * *p* < 0.05, *** *p* < 0.001, **** *p* < 0.0001, ns: not significant.

**Figure 2 ijms-23-14025-f002:**
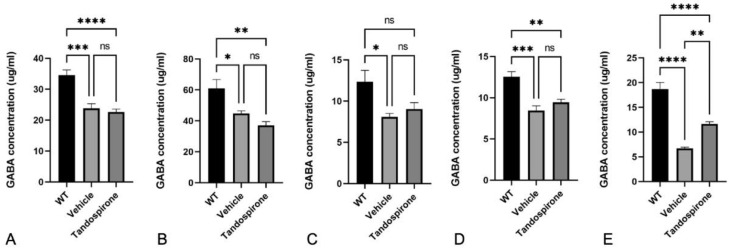
Effect of tandospirone treatment on GABA concentration in the cortex (**A**), midbrain (**B**), hippocampus (**C**), cerebellum (**D**) and brainstem (**E**). *Mecp2*-KO mice showed a significantly decreased GABA concentration in all brain regions, but administration of tandospirone only significantly increased GABA concentration in the brainstem. All values are expressed as the means ± SEMs. Statistical analysis was performed using one-way analysis of variance (ANOVA) with Tukey’s multiple comparison post hoc test for intergroup comparisons. *n* = 6, * *p* < 0.05, ** *p* < 0.01, *** *p* < 0.001, **** *p* < 0.0001, ns: not significant.

**Figure 3 ijms-23-14025-f003:**
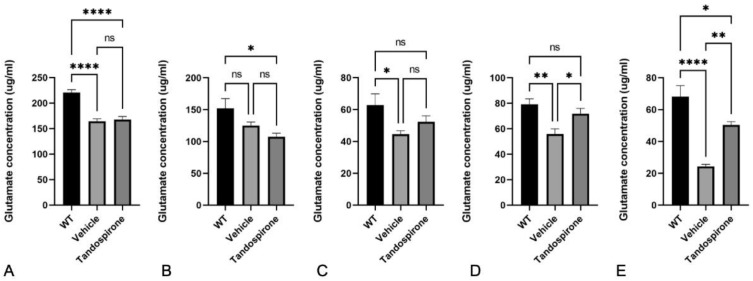
Effect of tandospirone treatment on glutamate concentration in the cortex (**A**), midbrain (**B**), hippocampus (**C**), cerebellum (**D**) and brainstem (**E**). *Mecp2*-KO mice showed a significantly decreased glutamate concentration in the cortex, hippocampus, cerebellum and brainstem when compared with WT mice. Administration of tandospirone significantly increased the lower glutamate concentration in the cerebellum and brainstem but no significant changes in the cortex, midbrain or hippocampus were observed. All values are expressed as the means ± SEMs. Statistical analysis was performed using one-way analysis of variance (ANOVA) with Tukey’s multiple comparison post hoc test for intergroup comparisons. *n* = 6, * *p* < 0.05, ** *p* < 0.01, **** *p* < 0.0001, ns: not significant.

**Figure 4 ijms-23-14025-f004:**
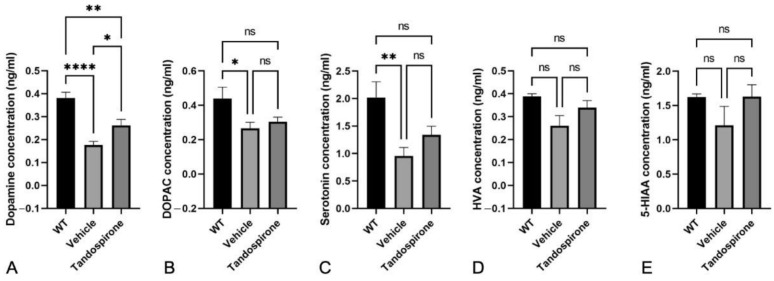
Effect of tandospirone treatment on dopamine, serotonin and their metabolites in the cerebellum. *Mecp2*-KO mice showed significantly lower dopamine (**A**), DOPAC (**B**) and serotonin (**C**) concentrations than WT mice in the cerebellum, but the differences are not significant for HVA (**D**) and 5-HIAA (**E**). Tandospirone treatment only significantly increased the lower dopamine content in the cerebellum of *Mecp2*-KO mice. All values are expressed as the means ± SEMs. Statistical analysis was performed using one-way analysis of variance (ANOVA) with Tukey’s multiple comparison post hoc test for intergroup comparisons. *n* = 6, * *p* < 0.05, ** *p* < 0.01, **** *p* < 0.0001, ns: not significant.

**Figure 5 ijms-23-14025-f005:**
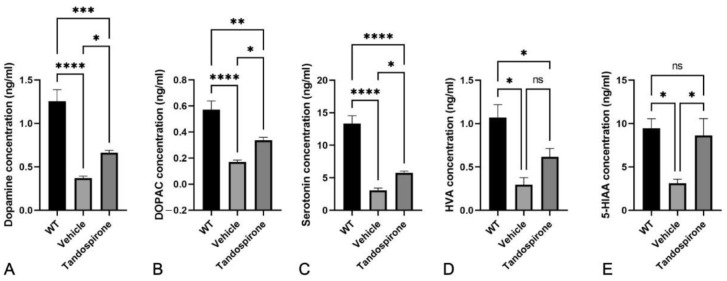
Effect of tandospirone treatment on dopamine, serotonin and their metabolites in the brainstem. *Mecp2*-KO mice showed significantly lower dopamine (**A**), DOPAC (**B**) and serotonin (**C**), HVA (**D**) and 5-HIAA (**E**) concentrations than WT mice in the brainstem. Tandospirone treatment significantly reversed the decrease in dopamine, DOPAC, serotonin and 5-HIAA in *Mecp2*-KO mice. All values are expressed as the means ± SEMs. Statistical analysis was performed using one-way analysis of variance (ANOVA) with Tukey’s multiple comparison post hoc test for intergroup comparisons. *n* = 6, * *p* < 0.05, ** *p* < 0.01, *** *p* < 0.001, **** *p* < 0.0001, ns: not significant.

**Figure 6 ijms-23-14025-f006:**
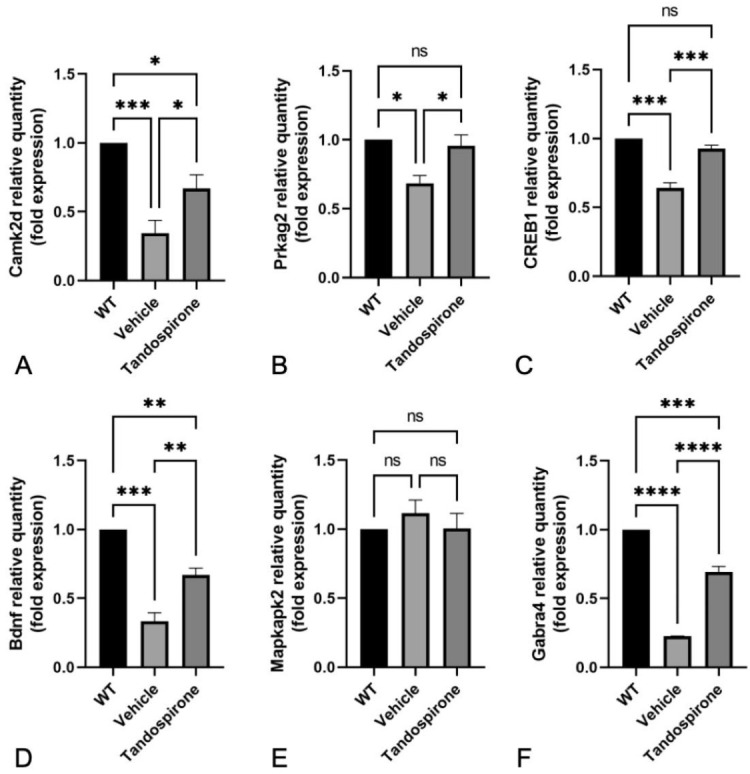
Gene expression in the midbrain by quantitative PCR. Expression changes in *CAMK2D* (**A**), *PRKAG2* (**B**), *CREB1* (**C**), *BDNF* (**D**), *MAPKAPK2* (**E**) and *GABRA4* (**F**) in WT, vehicle-treated and tandospirone-treated *Mecp2*-KO mice. All values are expressed as the means ± SEMs. Statistical analysis was performed using one-way analysis of variance (ANOVA) with Tukey’s multiple comparison post hoc test for intergroup comparisons. *n* = 4, * *p* < 0.05, ** *p* < 0.01, *** *p* < 0.001, **** *p* < 0.0001, ns: not significant.

## Data Availability

Data are available from the authors upon reasonable request and with permission from the Ethics Committee of the National Center of Neurology and Psychiatry.

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
