# Peer review of "5-HT1A Receptor Agonist Treatment Partially Ameliorates Rett Syndrome Phenotypes in mecp2-Null Mice by Rescuing Impairment of Neuron Transmission and the CREB/BDNF Signaling Pathway"

_ijms, 2022, doi:10.3390/ijms232214025_

Round 1

Reviewer 1 Report

This manuscript contributed by Dai and colleagues, titled “5-HT1A receptor agonist treatment partially ameliorates Rett syndrome 2 phenotypes in mecp2-null mice by rescuing impairment of neuron 3 transmission and the CREB/BDNF signaling pathway” The author uses a variety of techniques and KO mice to find tandospirone may reduce the Rett syndrome phenotypes, Then author proves the CREB1/BDNF signaling pathway involved in the mechanism. Overall, the author had a clear significance hypothesis and the experimental evidence was enough.

Some issues list below:

1 Dose the tandospirone dependent, why does the author use this in this paper?

2 Whether the tandospirone will affect the movement ability of animals, a rotarod test can help

Author Response

Response to Reviewer 1 Comments

We are glad to hear you. Here, according to your comments and suggestions, we corrected the text and figures. Please find them out.

Point 1: Dose the tandospirone dependent, why does the author use this in this paper?

Response 1: We decided the dose, according to the below reports (Miller et al., 1992; Akiyama et al.,1999). We preliminarily performed two high-dose study of 15mg/kg/day and 30mg/kg/day of tandospirone. As the results, the 30mg/kg/day dose did notextend the lifespan of Mecp2-KO mice, but significant in the dose of 15mg/kg/day. Therefore, this dose was used for this study. This reference is cited in the text.

Miller, L.G.; Thompson, M.L.; Byrnes, J.J.; Greenblatt, D.J.; Shemer, A. Kinetics, brain uptake, and receptor binding of tandospirone and its metabolite 1-(2-pyrimidinyl)-piperazine. J. Clin. Psychopharmacol. 1992, 12(5), 341-345.

Akiyama, M.; Kirihara, T.; Takahashi, S.; Minami, Y.; Yoshinobu, Y.; Moriya, T.; Shibata, S. Modulation of mPer1 gene expression by anxiolytic drugs in mouse cerebellum. Br. J. Pharmacol. 1999, 128(7), 1616-1622.

Point 2: Whether the tandospirone will affect the movement ability of animals, a rotarod test can help.

Response 2: Thanks for your suggestion. The evaluation of mobility we used has been often performed such as the below report. We think that tandospirone might affect the movement ability of animals. Many papers have reported that Mecp2-KO mouse is very weak and cannot be performed usual movement ability tests.  

Szczesna K, de la Caridad O, Petazzi P, Soler M, Roa L, Saez MA, Fourcade S, Pujol A, Artuch-Iriberri R, Molero-Luis M, Vidal A, Huertas D, Esteller M. Improvement of the Rett syndrome phenotype in a MeCP2 mouse model upon treatment with levodopa and a dopa-decarboxylase inhibitor. Neuropsychopharmacology. 2014, 39(12), 2846-2856. doi: 10.1038/npp.2014.136.

I am looking forward to hearing from you.

Best regards,

Masayuki Itoh

National Center of Neurology and Psychiatry

4-1-1 Ogawahigashi, Kodaira, Tokyo 1878502, Japan.

Reviewer 2 Report

The authors investigated the potential benefits of Tandospirone for the treatment of Rett syndrome (RTT). The effects of intraperitoneal administration of Tandospirone, have been evaluated in an animal model of RTT (Mecp2-null mice) and the results suggest that the treatment ameliorates some of the symptomatic features of RTT and extends animals’ survival. Authors detect variations of neurotransmitters and gene expression in specific brain areas, more specifically, hypothesize that the increase in BDNF signaling genes might be the key mechanism for the therapeutic potential of Tandospirone.

The paper is of interest in the field and some of the results shown (especially symptomatic assessment) are well-defined, however the correlation between Tandospirone and BDNF signaling could be investigated in more depth or the reasoning supporting this correlation could be implemented in the Discussion.

Please find below specific comments:

1. Symptomatic evaluation of disease progression:

- The authors show behavioral analysis only at 8 weeks (end of treatment), please explain why the assessment was not performed at several time point along animals’ lifespan.

- In the method section please include the protocol number for ethical authorization.

- In Fig.S1 A WT line is missing in the chart.

- In Fig.S1 caption the animal group size is indicated as n = 12~30 (line 494), please clarify.

2. Levels of neurotransmitters:

- Line 139. In the text figures have been erroneously cited: Figure S2 refer to cortex while in the text is cited to describe results of midbrain analysis (shown in Figure S3).

- Line 141. In the text figure S3 has been erroneously cited: Figure S3 refer to midbrain while in the text is cited to describe results of cortex analysis (shown in Figure S2).

- Line 152. Reduction of HVA and 5-HIAA are erroneously cited: the reduction is significant in Fig.S2D, S2E and S3D but NOT in S3E.

3. BDNF signaling

Tandospirone seems to increase several elements involved in the BDNF signaling at the transcriptional level, however, to substantiate this result additional analysis could be performed (such as for example western blot or immunohistochemistry) to show that the protein levels are increased as well. If this is not possible, at least include a comment in the discussion pointing at this limit of the analysis.  

Author Response

Response to Reviewer 2 Comments

We are glad to hear you. Here, according to your comments and suggestions, we corrected the text and figures. Please find them out.

Point 1: Symptomatic evaluation of disease progression: - The authors show behavioral analysis only at 8 weeks (end of treatment), please explain why the assessment was not performed at several time point along animals’ lifespan.

Response 1: Since the most animal models of Rett syndrome are characterized by onset of disease from 4 weeks and debilitation and death around 8 weeks, we chose to administer the treatment from 4 weeks to the end of 8 weeks and then observe the change of animals’ lifespan.

- In the method section please include the protocol number for ethical authorization.

Response: The number is 2020010 of the ethical committee of animal experiment of the institute and added in test.

- In Fig.S1 A WT line is missing in the chart.

Response: WT line is added in Fig. S1.

- In Fig.S1 caption the animal group size is indicated as n = 12~30 (line494), please clarify.

Response: vehicle: n=30, tandospirone: n=12. The numbers are added in the legend of Fig. S1.

Point 2: Levels of neurotransmitters: - Line 139. In the text figures have been erroneously cited: Figure S2 refer to cortex while in the text is cited to describe results of midbrain analysis (shown in Figure S3). - Line 141. In the text figure S3 has been erroneously cited: Figure S3 refer to midbrain while in the text is cited to describe results of cortex analysis (shown in Figure S2). - Line 152. Reduction of HVA and 5-HIAA are erroneously cited: the reduction is significant in Fig.S2D, S2E and S3D but NOT in S3E.

Response 2: We have corrected these mistakes.

Point 3: BDNF signaling Tandospirone seems to increase several elements involved in the BDNF signaling at the transcriptional level, however, to substantiate this result additional analysis could be performed (such as for example western blot or immunohistochemistry) to show that the protein levels are increased as well. If this is not possible, at least include a comment in the discussion pointing at this limit of the analysis.

Response 3: Thanks for your suggestion. Accordingly, we have added a comment in the discussion pointing at this limit of analysis.

I am looking forward to hearing from you.

Best regards,

Masayuki Itoh

National Center of Neurology and Psychiatry

4-1-1 Ogawahigashi, Kodaira, Tokyo 1878502, Japan.

Reviewer 3 Report

In this study, authors used a Mecp2-  null (KO) mouse model of RTT to investigate tandospirone treatment for the RTT phenotype. They show administration of tandospirone significantly extended the lifespan, ameliorated RTT phenotypes, hindlimb clasping, gait, tremor and breathing in Mecp2-KO mice. Moreover, authors also show that Tandospirone treatment significantly improved the impairment in GABAergic, glutaminergic, dopaminergic and serotoninergic neurotransmission in the brainstem. Additionally, authors performed RNA-seq analysis found that tandospirone modulates the RTT phenotype partially through the CREB1/BDNF signaling pathway in Mecp2-KO mice. Finally, the authors suggest a novel therapy for clinical treatment. Overall, this is a very interesting study. The manuscript is well written, and the data is well presented.

I recommend it for publication.

I have a minor comment

1)     I would recommend writing elaborative conclusive paragraph of the novelty of the study at the end of the introduction.

Author Response

Response to Reviewer 3 Comments

We are glad to hear you. Here, according to your comments and suggestions, we corrected the text and figures. Please find them out.

Point 1: I would recommend writing elaborative conclusive paragraph of the novelty of the study at the end of the introduction.

Response 1: Thank you for the recommendation, an elaborative conclusive paragraph of the novelty of the study has been added at the end of the introduction.

I am looking forward to hearing from you.

Best regards,

Masayuki Itoh

National Center of Neurology and Psychiatry

4-1-1 Ogawahigashi, Kodaira, Tokyo 1878502, Japan.
